# Causal Confirmation Measures: From Simpson’s Paradox to COVID-19

**DOI:** 10.3390/e25010143

**Published:** 2023-01-10

**Authors:** Chenguang Lu

**Affiliations:** 1Intelligence Engineering and Mathematics Institute, Liaoning Technical University, Fuxin 123000, China; survival99@gmail.com; 2School of Computer Engineering and Applied Mathematics, Changsha University, Changsha 410022, China

**Keywords:** causal confirmation, Bayesian confirmation, causal inference, semantic information measure, cross-entropy, Simpson’s Paradox, COVID-19, risk measures

## Abstract

When we compare the influences of two causes on an outcome, if the conclusion from every group is against that from the conflation, we think there is Simpson’s Paradox. The Existing Causal Inference Theory (ECIT) can make the overall conclusion consistent with the grouping conclusion by removing the confounder’s influence to eliminate the paradox. The ECIT uses relative risk difference *P_d_* = max(0, (*R* − 1)/*R*) (*R* denotes the risk ratio) as the probability of causation. In contrast, Philosopher Fitelson uses confirmation measure *D* (posterior probability minus prior probability) to measure the strength of causation. Fitelson concludes that from the perspective of Bayesian confirmation, we should directly accept the overall conclusion without considering the paradox. The author proposed a Bayesian confirmation measure *b** similar to *P_d_* before. To overcome the contradiction between the ECIT and Bayesian confirmation, the author uses the semantic information method with the minimum cross-entropy criterion to deduce causal confirmation measure *Cc* = (*R* − 1)/max(*R*, 1). *Cc* is like *P_d_* but has normalizing property (between −1 and 1) and cause symmetry. It especially fits cases where a cause restrains an outcome, such as the COVID-19 vaccine controlling the infection. Some examples (about kidney stone treatments and COVID-19) reveal that *P_d_* and *Cc* are more reasonable than *D*; *Cc* is more useful than *P_d_*.

## 1. Introduction

Causal confirmation is the expansion of Bayesian confirmation. It is also a task of causal inference. The Existing Causal Inference Theory (ECIT), including Rubin’s (or Neyman-Rubin) potential outcomes model [1,2] and Pearl’s causal graph [3,4], has achieved great success. However, causal confirmation is rarely mentioned.

Bayesian confirmation theories are also called confirmation theories, which can be divided into incremental and inductive schools. The incremental school affirms that the confirmation measures the supporting strength of evidence *e* to hypothesis *h*, as explained by Fitelson [5]. Following Carnap [6], the incremental school’s researchers often use the increment of a hypothesis’ probability or logical probability, *P*(*h|e*) − *P*(*h*), as a confirmation measure. Fitelson [5] discussed causal confirmation with this measure and obtained some conclusions incompatible with the ECIT. On the other hand, the inductive school [7,8] considers confirmation as induction’s modern form, whose task is to measure a major premise’s creditability supported by a sample or sampling distribution.

We use *e*→*h* to denote a major premise. Variable *e* takes one of two possible values *e*_1_ and its negation *e*_0._ Variable *h* takes one of two possible values *h*_1_ and its negation *h*_0_. Then a sample includes four examples (*e*_1_, *h*_1_), (*e*_1_, *h*_0_), (*e*_0_, *h*_1_), and (*e*_0_, *h*_0_) with different proportions. The inductive school’s researchers often use positive examples and counterexamples’ proportions (*P*(*e*_1_*|h*_1_) and *P*(*e*_1_*|h_0_*)) or likelihood ratio (*P*(*e*_1_*|h*_1_)*/P*(*e*_1_*|h*_0_)) to express confirmation measures.

A confirmation measure is often denoted by *C*(*e, h*) or *C*(*h, e*). The author (of this paper) agrees with the inductive school and suggests using *C*(*e*→*h*) to express a confirmation measure so that the task is evident [8]. In this paper, we use “*x=*>*y*” to denote “Cause *x* leads to outcome *y*”.

Although the two schools understand confirmation differently, both use sampling distribution *P*(*e*, *h*) to construct confirmation measures. There have been many confirmation measures [8,9]. Most researchers agree that an ideal confirmation measure should have the following two desired properties:
normalizing property [9,10], which means *C*(*e*, *h*) should change between −1 and 1 so that the difference between a rule *e*→*h* and the best or the worst rule is clear;hypothesis symmetry [11] or consequent symmetry [8], which means *C*(*e*_1_→*h*_1_) = −*C*(*e*_1_→*h*_0_). For example, *C*(raven→black) = −*C*(raven→non-black).


The author in [8] distinguished channels’ confirmation and predictions’ confirmation and provided channels’ confirmation measure *b**(*e→h*) and predictions’ confirmation measure *c**(*e*→*h*). Both have the above two desired properties and can be used for the probability predictions of *h* according to *e*.

Bayesian confirmation confirms associated relationships, which are different from causal relationships. Association includes causality, but many associated relationships are not causal relationships. One reason is that the existence of association is symmetrical (if *P*(*h|e*) ≠ 0, then *P*(*e|h*) ≠ 0), whereas the existence of causality is asymmetrical. For example, in medical tests, *P*(positive|infected) reflects both association and causality. However, inversely, *P*(infected|positive) only indicates association. Another reason is that two associated events, *A* and *B*, such as electric fans’ easy selling and air conditioners’ easy selling, are the outcomes caused by the third event (hot weather). Neither *P*(*A|B*) nor *P*(*B|A*) indicates causality.

Causal inference only deals with uncertain causal relationships in nature and human society without considering those in mathematics, such as (*x* + 1)(*x* − 1) < *x*^2^ because (*x* + 1)(*x* − 1) = *x*^2^ − 1. We know that Kant distinguishes analytic judgments and synthetic judgments. Although causal inference is a mathematical method, it is used for synthetic judgments to obtain uncertain rules in biology, psychology, economics, etc. In addition, causal confirmation only deals with binary causality.

Although causal confirmation was rarely mentioned in the ECIT, the researchers of causal inference and epidemiology have provided many measures (without using the term “confirmation measure”) to indicate the strength of causation. These measures include risk difference [12]:(1)$$RD=P(y_{1}|x_{1})-P(y_{1}|x_{0}),$$
relative risk difference or the risk ratio (like the likelihood ratio for medical tests):(2)$$RR=P(y_{1}|x_{1})/P(y_{1}|x_{0}),$$
and the probability of causation *P_d_* (used by Rubin and Greenland [13]) or the probability of necessity *PN* (used by Pearl [3]). There is:(3)$$P_{d}=PN=\max(0,\frac{P(y_{1}|x_{1})-P(y_{1}|x_{0})}{P(y_{1}|x_{1})}).$$
*P_d_* is also called Relative Risk Reduction (RRR) [12]. In the above formula, max(0, ∙) means its minimum is 0. This function is to make *P_d_* more like a probability. Measure *b** proposed by the author in [8] is like *P_d_*, but *b** changes between −1 and 1. The above risk measures can measure not only risk or relative risk but also success or relative success raised by the cause.

The risk measures in Equations (1)–(3) are significant; however, they do not possess the two desired properties and hence are improper as causal confirmation measures.

We will encounter Simpson’s Paradox if we only use sampling distributions for the above measures. Simpson’s Paradox has been accompanying the study of causal inference, as the Raven Paradox has been going with the study of Bayesian confirmation. Simpson proposed the paradox [14] using the following example.

**Example** **1** [15]**.** *The admission data of the graduate school of the University of California, Berkeley (UCB), for the fall of 1973 showed that 44% of male applicants were accepted, whereas only 35% of female applicants were accepted. There was probably gender bias present. However, in most departments, female applicants’ acceptance rates were higher than male applicants*.

Was there a gender bias? Should we accept the overall conclusion or the grouping conclusion (i.e., that from every department)? If we take the overall conclusion, we can think that the admission had a bias against the female. On the other hand, if we accept the grouping conclusion, we can say that the female applicants were priorly accepted. Therefore, we say there exists a paradox.

Example 1 is a little complicated and easy to raise arguments against. To simplify the problem, we use Example 2, which the researchers of causal inference often mentioned, to explain Simpson’s Paradox quantitatively.

We use *x*_1_ to denote a new cause (or treatment) and *x*_0_ to denote a default cause or no cause. If we need to compare two causes, we may use *x*_1_ and *x*_2_, or *x_i_* and *x_j_*, to represent them. In these cases, we may assume that one is default like *x*_0_.

**Example** **2** [16,17]**.** *Suppose there are two treatments, x_1_ and x_2_, for patients with kidney stones. Patients are divided into two groups according to their size of stones. Group g_1_ includes patients with small stones, and group g_2_ has large ones. Outcome y_1_ represents the treatment’s success. Success rates shown in Figure 1 are possible. In each group, the success rate of x_2_ is higher than that of x_1_; however, the overall conclusion is the opposite*.

According to Rubin’s potential outcomes model [1], we should accept the grouping conclusion: *x*_2_ is better than *x*_1_. The reason is that the stones’ size is a confounder, and the overall conclusion is affected by the confounder. We should eliminate this influence. The method is to imagine the patients’ numbers in each group are unchanged whether we use *x*_1_ or *x*_2._ Then we replace weighting coefficients *P*(*g_i_*|*x*_1_) and *P*(*g_i_|x*_2_) with *P*(*g_i_*) (*i* = 1, 2) to obtain two new overall success rates. Rubin [1] expresses them as *P*(*y*_1_*^x^*^1^) and *P*(*y*_1_*^x^*^2^); whereas Pearl [3] expresses them as *P*(*y*_1_|do(*x*_1_)) and *P*(*y*_1_|do(*x*_2_)). Then, the overall conclusion is consistent with the grouping conclusion.

Should we always accept the grouping conclusion when the two conclusions are inconsistent? It is not sure! Example 3 is a counterexample.

**Example** **3** (from [18])**.** *Treatment x_1_ denotes taking a kind of antihypertensive drug, and treatment x_0_ means taking nothing. Outcome y_1_ denotes recovering health, and y_0_ means not. Patients are divided into group g_1_ (with high blood pressure) and group g_0_ (with low blood pressure). It is very possible that in each group g, P(y_1_|g, x_1_) < P(y_1_|g, x_0_) (which means x_0_ is better than x_1_); whereas overall result is P(y_1_|x_1_) > P(y_1_|x_0_) (which means x_1_ is better than x_0_).*

The ECIT tells us that we should accept the overall conclusion that *x*_1_ is better than *x*_0_ because blood pressure is a mediator, which is also affected by *x*_1_. We expect that *x*_1_ can move a patient from *g*_1_ to *g*_0_; hence we need not change the weighting coefficients from *P*(*g*|*x*) to *P*(*g*). The grouping conclusion, *P*(*y*_1_|*g*, *x*_1_) < *P*(*y*_1_|*g*, *x*_0_), exists because the drug has a side effect.

There are also some examples where the grouping conclusion is acceptable from one perspective, and the overall conclusion is acceptable from another.

**Example** **4** [19]**.** *The United States statistical data about COVID-19 in June 2020 show that COVID-19 led to a higher Case Fatality Rate (CFR) of non-Hispanic whites than others (overall conclusion). We can find that only 35.3% of the infected people were non-Hispanic whites, whereas 49.5% of the infected people who died from COVID-19 were non-Hispanic whites. It seems that COVID-19 is more dangerous to non-Hispanic whites. However, Dana Mackenzie pointed out* [19] *that we will obtain the opposite conclusion from every age group because the CFR of non-Hispanic whites is lower than that of other people in every age group. So, there exists Simpson’s Paradox. The reason is that non-Hispanic whites have longer lifespans and a relatively large proportion of the elderly, while COVID-19 is more dangerous to the elderly*.

Kügelgen et al. [20] also pointed out the existence of Simpson’s Paradox after they compared the CFRs of COVID-19 (reported in 2020) in China and Italy. Although the overall conclusion was that the CFR in Italy was higher than in China, the CFR of every age group in China was higher than in Italy. The reason is that the proportion of the elderly in Italy is larger than in China.

According to Rubin’s potential outcomes model or Pearl’s causal graph, if we think that the reason for non-Hispanic whites’ longevity is good medical conditions or other elements instead of their race, then the lifespan is a confounder. Therefore, we should accept the grouping conclusion. On the other hand, if we believe that non-Hispanic whites are longevous because they are whites, then the lifespan is a mediator, so we should accept the overall conclusion.

Example 1 is similar to Example 4, but the former is not easy to understand. The data show that the female applicants tended to choose majors with low admission rates (perhaps because lower thresholds resulted in more intense competition). This tendency is like the lifespan of the white. If we regard lifespan as a confounder, Berkeley University had no gender bias against the female. On the other hand, if we believe the female is the cause of this tendency, the overall conclusion is acceptable, and gender bias should have existed. Deciding which of the two judgments is right depends on one’s perspective.

Pearl’s causal graph [3] makes it clear that for the same data, if supposed causal relationships are different, conclusions are also different. So, it is not enough to have data only. We also need the structural causal model.

However, the incremental school’s philosopher Fitelson argues that from the perspective of Bayesian confirmation, we should accept the overall conclusion according to the data without considering causation; Simpson’s Paradox does not exist according to his rational explanation. His reason is that we can use the measure [5]:*measure _i_* = *P*(*y*_1_|*x*_1_) − *P*(*y*_1_)(4)
to measure causality. Fitelson proves (see Fact 3 of Appendix in [5]) that if there is:*P*(*y*_1_|*x*_1_, *g_i_*) > *P*(*y*_1_*|x*_2_, *g_i_*), *i*=1, 2, (5)
then there must be *P*(*y*_1_|*x*_1_) > *P*(*y*_1_). The result is the same when “>“ is replaced with “<“. Therefore, Fitelson affirms that, unlike *RD* and *P_d_*, *measure i* does not result in the paradox.

However, Equation (5) expresses a rigorous condition, which excludes all examples with joint distributions *P*(*y*, *x*, *g*) that cause the paradox, including Fitelson’s simplified example about the admissions of the UCB.

One cannot help asking:For Example 2 about kidney stones, is it reasonable to accept the overall conclusion without considering the difficulties of treatments?Is it necessary to extend or apply a Bayesian confirmation measure incompatible with the ECIT and medical practices to causal confirmation?Except for the incompatible confirmation measures, are there no compatible confirmation measures?

In addition to the incremental school’s confirmation measures, there are also the inductive school’s confirmation measures, such as *F* proposed by Kemeny and Oppenheim in 1952 and *b** provided by the author in 2020.

This paper mainly aims at:combining the ECIT to deduce causal confirmation measure *Cc*(*x*_1_ => *y*_1_) (“*C*” stands for confirmation and “*c*” for the cause), which is similar to *P_d_* but can measure negative causal relationships, such as “vaccine => infection”;explaining that measures *Cc* and *P_d_* are more suitable for causal confirmation than *measure _i_* by using some examples with Simpson’s Paradox;supporting the inductive school of Bayesian confirmation in turn.

When the author proposed measure *b**, he also provided measure *c** for eliminating the Raven Paradox [8]. For extending *c** to causal confirmation, this paper presents measure *Ce*(*x*_1_ => *y*_1_), which indicates the outcome’s inevitability or the cause’s sufficiency.

## 2. Background

### 2.1. Bayesian Confirmation: Incremental School and Inductive School

A universal judgment is equivalent to a hypothetical judgment or a rule, such as “All ravens are black” is equivalent to “For every *x*, if *x* is a raven, then *x* is black”. Both can be used as a major premise for a syllogism. Due to the criticism of Hume and Popper, most philosophers no longer expect to obtain absolutely correct universal judgments or major premises by induction but hope to obtain their degrees of belief. A degree of belief supported by a sample or sampling distribution is the degree of confirmation.

It is worth noting that a proposition does not need confirmation. Its truth value comes from its usage or definition [8]. For example, “People over 18 are adults” does not need confirmation; whether it is correct depends on the government’s definition. Only major premises (such as “All ravens are black” and “If a person’s Nucleic Acid Test is positive, he is likely to be infected with COVID-19”) need confirmation.

A natural idea is to use conditional probability *P*(*h|e*) to confirm a major premise or rule denoted with *e*→*h*. This measure is also recommended by Fitelson [5], and called *confirm _f_*. There is [5,6]:*confirm _f_* = *f*(*e,h*) = *P*(*h|e*). (Carnap, 1962, Fitelson, 2017)

However, *P*(*h|e*) depends very much on the prior probability *P*(*h*) of *h*. For example, where COVID-19 is prevalent, *P*(*h*) is large, and *P*(*h|e*) is also large. Therefore, *P*(*h|e*) cannot reflect the necessity of *e*. An extreme example is that *h* and *e* are independent of each other, but if *P*(*h*) is large, *P*(*h|e*) = *P*(*h, e*)/*P*(*e*) = *P*(*h*) is also large. At this time, *P*(*h|e*) does not reflect the creditability of the causal relationship. For example, *h* = “There will be no earthquake tomorrow”, *P*(*h*) = 0.999, and *e* = “Grapes are ripe”. Although *e* and *h* are irrelative, *P*(*h|e*) = *P*(*h*) = 0.999 is very large. However, we cannot say that the ripe grape supports no earthquake happening.

For this reason, the incremental school’s researchers use posterior (or conditional) probability minus prior probability to express the degree of confirmation. These confirmation measures include [6,10,21,22]:*D*(*e*_1_, *h*_1_) = *P*(*h*_1_|*e*_1_) − *P*(*h*_1_)     (Carnap, 1962),
*M*(*e*_1_, *h*_1_) = *P*(*e*_1_|*h*_1_) − *P*(*e*_1_)     (Mortimer, 1988),
*R*(*e*_1_, *h*_1_) = log[*P*(*h*_1_|*e*_1_)/*P*(*h*_1_)]     (Horwich, 1982),
*C*(*e*_1_, *h*_1_) = *P*(*h*_1_, *e*_1_) − *P*(*e*_1_) *P*(*h*_1_)     (Carnap, 1962),
$$Z(h_{1},e_{1})=\{\begin{array}{l} {}[P(h_{1}|e_{1})-P(h_{1})]/P(h_{0},\;\mathrm{as}\;P(h_{1}|e_{1})\geq P(h_{1}),\\ {}[P(h_{1}|e_{1})-P(h_{1})]/P(h_{1}),\;\mathrm{otherwise}, \end{array}\;(\mathrm{Crupi}\;\mathrm{et}\;\mathrm{al}.,\;2007).$$ In the above measures, *D*(*e*_1_, *h*_1_) is *measure _i_* recommended by Fitelson in [5]. *R*(*e*_1_, *h*_1_) is an information measure. It can be written as log*P*(*h*_1_|*e*_1_) − log*P*(*h*_1_). Since log*P*(*h*_1_|*e*_1_) − log*P*(*h*_1_) = log*P*(*e*_1_|*h*_1_) − log*P*(*e*_1_) = log*P*(*h*_1_,*e*_1_) − log[*P*(*h*_1_)*P*(*e*_1_)], *D*, *M*, and *C* increase with *R* and hence can be replaced with each other. *Z* is the normalization of *D* for having the two desired properties [10]. Therefore, we can also call the incremental school the information school.

On the other hand, the inductive school’s researchers use the difference (or likelihood ratio) between two conditional probabilities representing the proportions of positive and negative examples to express confirmation measures. These measures include [7,8,23,24,25]:S(*e*_1_, *h*_1_) = *P*(*h*_1_|*e*_1_) − *P*(*h*_1_|*e*_0_)     (Christensen, 1999),
*N*(*e*_1_, *h*_1_) = *P*(*e*_1_|*h*_1_) − *P*(*e*_1_|*h*_0_)     (Nozick, 1981),
*L*(*e*_1_, *h*_1_) = log[ *P*(*e*_1_|*h*_1_)/*P*(*e*_1_|*h*_0_)]     (Good, 1984),
*F*(*e*_1_, *h*_1_) = [*P*(*e*_1_|*h*_1_) − *P*(*e*_1_|*h*_0_)]/[ *P*(*e*_1_|*h*_1_)+ *P*(*e*_1_|*h*_0_)] (Kemeny and Oppenheim, 1952),
*b**(*e*_1_, *h*_1_) = [*P*(*e*_1_|*h*_1_) − *P*(*e*_1_|*h*_0_)]/max(*P*(*e*_1_|*h*_1_), *P*(*e*_1_|*h*_0_)) (Lu, 2020).

They are all positively related to the Likelihood Ratio (*LR*^+^ = *P*(*e*_1_|*h*_1_)/*P*(*e*_1_|*h*_0_)). For example, *L* = log *LR*^+^ and *F* = (*LR*^+^ − 1)/(*LR*^+^ + 1) [7]. Therefore, these measures are compatible with risk (or reliability) measures, such as *P_d_*, used in medical tests and disease control. Although the author has studied semantic information theory for a long time [26,27,28] and believe both schools have made important contributions to Bayesian confirmation, he is on the side of the inductive school. The reason is that information evaluation occurs before classification, whereas confirmation is needed after classification [8].

Although the researchers understand confirmation differently, they all agree to use a sample including four types of examples (*e*_1_, *h*_1_), (*e*_0_, *h*_1_), (*e*_1_, *h*_0_), and (*e*_0_, *h*_0_) with different proportions as the evidence to construct confirmation measures [8,10]. The main problem with the incremental school is that they do not distinguish the evidence of a major premise and that of the consequent of the major premise well. When they use the four examples’ proportions to construct confirmation measures, *e* is regarded as the major premise’s antecedent, whose negation *e*_0_ is meaningful. However, when they say “to evaluate the supporting strength of *e* to *h*”, *e* is understood as a sample, whose negation *e*_0_ is meaningless. It is more meaningless to put a sample *e* or *e*_0_ in an example (*e*_1_, *h*_1_) or (*e*_0_, *h*_1_).

We compare *D* (i.e., *measure _i_*) and *S* to show the main difference between the two schools’ measures. Since:*D*(*e*_1_, *h*_1_) = *P*(*h*_1_|*e*_1_) − *P*(*h*_1_) = *P*(*h*_1_|*e*_1_) − [*P*(*e*_1_)*P*(*h*_1_|*e*_1_) + *P*(*e*_0_)*P*(*h*_1_|*e*_0_)] = [1 − *P*(*e*_1_)]*P*(*h*_1_|*e*_1_) − *P*(*e*_0_)*P*(*h*_1_|*e*_0_) = *P*(*e*_0_)*S*(*e*_1_,*h*_1_),(6)
we can find that *D* changes with *P*(*e*_0_) or *P*(*e*_1_), but *S* does not. *P*(*e*) means the source and *P*(*h|e*) means the channel. *D* is related to the source and the channel, but *S* is only related to the channel. Measures *F* and *b** are also only related to channel *P*(*e|h*). Therefore, the author calls *b** the channels’ confirmation measure.

### 2.2. The P-T Probability Framework and the Methods of Semantic Information and Cross-Entropy for Channels’ Confirmation Measure b*(e→h)

In the P-T probability framework [28] proposed by the author, there are both statistical probability *P* and logical probability (or truth value) *T*; the truth function of a predicate is also a membership function of a fuzzy set [29]. Therefore, the truth function also changes between 0 and 1. The purpose of proposing this probability framework is to set up the bridge between statistics and fuzzy logic.

Let *X* be a random variable representing an instance, taking a value *x*∈*A* = {*x*_0_,*x*_1_,…}, and *Y* be a random variable representing a label or hypothesis, taking a value *y*∈*B* = { *y*_0_,*y*_1_,…}. The Shannon channel is a conditional probability matrix *P*(*y*_j_|*x_i_*) (*i* = 1,2,...; *j* = 1,2,…) or a set of transition probability functions *P*(*y*_j_|*x*) (*j* = 1,2,…). The semantic channel is a truth value matrix *T*(*y_j_|x_i_*) (*i* = 1,2,…; *j* = 1,2,…) or a set of truth functions *T*(*y*_j_|*x*) (*j* = 0,1,…). Let the elements in *A* that make *y_j_* true form a fuzzy subset *θ_j_*. The membership function *T*(*θ_j_*|*x*) of *θ_j_* is also the truth function *T*(*y*_j_|*x*) of *y_j_*, i.e., *T*(*θ_j_*|*x*) = *T*(*y_j_*|*x*).

The logical probability of *y_j_* is:(7)$$T(y_{j})=T(\theta_{j})=\sum_{i}P(x_{i})T(\theta_{j}|x_{i}).$$ Zadeh calls it the fuzzy event’s probability [30]. When *y_j_* is true, the conditional probability of *x* is:(8)$$P(x|\theta_{j})=P(x)T(\theta_{j}|x)/T(\theta_{j}).$$

Fuzzy set *θ_j_* can also be understood as a model parameter; hence *P*(*x*|*θ_j_*) is a likelihood function.

The differences between logical probability and statistical probability are:The statistical probability is normalized (the sum is 1), whereas the logical probability is not. Generally, we have *T*(*θ*_0_) + *T*(*θ*_1_) + … > 1.The maximum value of *T*(*θ_j_*|*x*) is 1 for different *x*, whereas *P*(*y*_0_|*x*) + *P*(*y*_1_|*x*) + … = 1 for a given *x*.

We can use the sample distribution to optimize the model parameters. For example, we use *x* to represent the age, use a logistic function as the truth function of the elderly: *T*(“elderly”|*x*) = 1/[1 + exp (− *bx + a*)], and use a sampling distribution to optimize *a* and *b*.

The (amount of) semantic information about *x_i_* conveyed by *y_j_* is:(9)$$I(x_{i};\theta_{j})=\log\frac{P(x_{i}|\theta_{j})}{P(x_{i})}=\log\frac{T(\theta_{j}|x_{i})}{T(\theta_{j})}.$$ For different *x*, the average semantic information conveyed by *y_j_* is:(10)$$\begin{array}{ll} I(X;\theta_{j}) & =\sum_{i}P(x_{i}|y_{j})\log\frac{T(\theta_{j}|x_{i})}{T(\theta_{j})}=\sum_{i}P(x_{i}|y_{j})\log\frac{P(x_{i}|\theta_{j})}{P(x_{i})}\\ & =-\sum_{i}P(x_{i}|y_{j})\log P(x_{i})-H(X|\theta_{j}). \end{array}$$ In the above formula, *H(X|θ_j_*) is a cross-entropy:(11)$$H(X|\theta_{j})=-\sum_{i}P(x_{i}|y_{j})\log P(x_{i}|\theta_{j}).\;$$ The cross-entropy has an important property: when we change *P*(*x*|*θ_j_*) so that *P*(*x*|*θ_j_*) = *P*(*x|y_j_*), *H(X|θ_j_*) reaches its minimum. It is easy to find from Equation (10) that *I(X*; *θ_j_*) reaches its maximum as *H(X|θ_j_*) reaches its minimum. The author has proved that if *P*(*x*|*θ_j_*) = *P*(*x|y_j_*), then *T*(*θ_j_*|*x*)∝*P*(*y*_j_|*x*) [27]. If for all *j*, *T*(*θ_j_*|*x*)∝*P*(*y*_j_|*x*), we say that the semantic channel matches the Shannon channel.

We use the medical test as an example to deduce the channels’ conformation measure *b**. We define *h*∈{*h*_0_, *h_1_*} = {infected, uninfected} and *e*∈{*e*_0_, *e*_1_} = {positive, negative}. The Shannon channel is *P*(*e|h*), and the semantic channel is *T*(*e|h*). The major premise to be confirmed is *e*_1_→*h*_1_, which means “If one’s test is positive, then he is infected”.

We regard a fuzzy predicate *e*_1_(*h*) as the linear combination of a clear predicate (whose truth value is 0 or 1) and a tautology (whose truth value is always 1). Let the tautology’s proportion be *b*_1_′ and the clear predicate’s proportion be 1 − *b*_1_′. Then we have:*T*(*e*_1_|*h*_0_) = *b*_1_′; *T*(*e*_1_|*h*_1_) = *b*_1′_+ *b*_1_ = *b*_1_′ + (1 − *b*_1_′) = 1. (12)

The *b*_1_′ is also called the degree of disbelief of rule *e*_1_→*h*_1_. The degree of disbelief optimized by a sample, denoted by *b*_1_′*, is the degree of disconfirmation. Let *b*_1_* denote the degree of confirmation; we have *b*_1_′* = 1 − |*b*_1_*|. By maximizing average semantic information *I*(*H*; *θ*_1_) or minimizing cross-entropy *H(H|θ_j_*), we can deduce (see Section 3.2 in [8]):(13)$${b_{1}}^{*}=b^{*}(e_{1}\to h_{1})=\frac{P(e_{1}|h_{1})-P(e_{1}|h_{0})}{\max(P(e_{1}|h_{1}),P(e_{1}|h_{0}))}=\frac{LR^{+}-1}{\max(LR^{+},1)}.$$

Suppose that likelihood function *P*(*h*|*e*_1_) is decomposed into an equiprobable part and a part with 0 and 1. Then, we can deduce the predictions’ confirmation measure *c**:(14)$${c_{1}}^{*}=c^{*}(e_{1}\to h_{1})=\frac{P(h_{1}|e_{1})-P(h_{0}|e_{1})}{\max(P(h_{1}|e_{1}),P(h_{0}|e_{1}))}=\frac{2P(h_{1}|e_{1})-1}{\max(P(h_{1}|e_{1}),1-P(h_{1}|e_{1}))}.$$

Measure *b** is compatible with the likelihood ratio and suitable for evaluating medical tests. In contrast, measure *c** is appropriate to assess the consequent inevitability of a rule and can be used to clarify the Raven Paradox [8]. Moreover, both measures have the normalizing property and symmetry mentioned above.

### 2.3. Causal Inference: Talking from Simpson’s Paradox

According to the ECIT, the grouping conclusion is acceptable for Example 2 (about kidney stones), whereas the overall conclusion is acceptable for Example 3 (about blood pressure). The reason is that *P*(*y*_1_|*x*_1_) and *P*(*y*_1_|*x*_0_) may not reflect causality well; in addition to the observed data or joint probability distribution *P*(*y*, *x*, *g*), we also need to suppose the causal structure behind the data [3].

Suppose there is the third variable, *u*. Figure 2 shows the causal relationships in Examples 2, 3, and 4. Figure 2a shows the causal structure of Example 2, where *u* (kidney stones’ size) is a confounder that affects both *x* and *y*. Figure 2b describes the causal structure of Example 3, where *u* (blood pressure) is a mediator that affects *y* but is affected by *x*. In Figure 2c, *u* can be interpreted as either a confounder or a mediator. The causality will differ from different perspectives, and *P*(*y*_1_|do(*x*)) will also differ. In all cases, we should replace *P*(*y*|*x*) with *P*(*y*|do(*x*)) (if they are different) to get *RD*, *RR*, and *P_d_*.

We should accept the overall conclusion for the example where *u* is a mediator. However, for the example where *u* is a confounder, how do we obtain a suitable *P*(*y*|do(*x*))? According to Rubin’s potential outcomes model, we use Figure 3 to explain the difference between *P*(*y*|do(*x*)) and *P*(*y*|*x*).

To find the difference in the outcomes caused by *x*_1_ and *x*_2_, we should compare the two outcomes in the same background. However, there is often no situation where other conditions remain unchanged except for the cause. For this reason, we need to replace *x*_1_ with *x*_2_ in our imagination and see the shift in *y*_1_ or its probability. If *u* is a confounder and not affected by *x*, the number of members in *g*_1_ and *g*_2_ should be unchanged with *x*, as shown in Figure 3. The solution is to use *P*(*g*) instead of *P*(*g*|*x*) for the weighting operation so that the overall conclusion is consistent with the grouping conclusion. Hence, the paradox no longer exists.

Although *P*(*x*_0_) + *P*(*x*_1_) = 1 is tenable, *P*(do (*x*_1_)) + *P*(do (*x*_0_)) = 1 is meaningless. That is why Rubin emphasizes that *P*(*y^x^*), i.e., *P*(*y*|do(*x*)), is still a marginal probability instead of a conditional probability, in essence.

Rubin’s reason [2] for replacing *P*(*g*|*x*) with *P*(*g*) is that for each group, such as *g*_1_, the two subgroups’ members (patients) treated by *x*_1_ and *x*_2_ are interchangeable (i.e., Pearl’s causal independence assumption mentioned in [5]). If a member is divided into the subgroup with *x*_1_, its success rate should be *P*(*y*_1_|*g*, *x*_1_); if it is divided into the subgroup with *x*_2_, the success rate should be *P*(*y*_1_|*g*, *x*_2_). *P*(*g*|*x*_1_) and *P*(*g|x*_2_) are different only because half of the data are missing. However, we can fill in the missing data using our imagination.

If *u* is a mediator, as shown in Figure 2b, a member in *g*_1_ may enter *g*_2_ because of *x*, and vice versa. *P*(*g*|*x*_0_) and *P*(*g*|*x*_1_) are hence different without needing to be replaced with *P*(*g*). We can let *P*(*y*_1_|do (*x*)) = *P*(*y*_1_|*x*) directly and accept the overall conclusion.

### 2.4. Probability Measures for Causation

In Rubin and Greenland’s article [13]:*P*(*t*) = [*R*(*t*) − 1]/*R*(*t*)(15)
is explained as the probability of causation, where *t* is one’s age of exposure to some harmful environment. *R*(*t*) is the age-specific infection rate (infected population divided by uninfected population). Let *y*_1_ stand for the infection, *x*_1_ for the exposure, and *x*_0_ for no exposure. Then there is *R*(*t*) = *P*(*y*_1_|do(*x*_1_), *t*)/*P*(*y*_1_|do(*x*_0_), *t*). Its lower limit is 0 because the probability cannot be negative. When the change of *t* is neglected, considering the lower limit, we can write the probability of causation as:(16)$$P_{d}=\max(0,\frac{R-1}{R})=\max(0,\frac{P(y_{1}|\mathrm{do}(x_{1}))-P(y_{1}|\mathrm{do}(x_{0}))}{P(y_{1}|\mathrm{do}(x_{1}))}).$$Pearl uses *PN* to represent *P_d_* and explains *PN* as the probability of necessity [3]. *P_d_* is very similar to confirmation measure *b** [8]. The main difference is that *b** changes between −1 and 1.

Robert van Rooij and Katrin Schulz [31] argue that conditionals of the form “If *x*, then *y*” are assertable only if:(17)$$\Delta^{*}P_{x}^{y}=\frac{P(y_{1}|x_{1})-P(y_{1}|x_{0})}{1-P(y_{1}|x_{0})}$$
is high. This measure is similar to confirmation measure *Z*. The difference between *P_d_* and Δ**P_x_^y^* is that *P_d_*, like *b**, is sensitive to counterexamples’ proportion *P*(*y*_1_|*x*_0_), whereas Δ**P_x_^y^* is not. Table 1 shows their differences.

David E. Over et al. [32] support the Ramsey test hypothesis, implying that the subjective probability of a natural language conditional, *P*(if *p* then *q*), is the conditional subjective probability, *P*(*q|p*). This measure is *confirm _f_* in [5].

The author [8] suggests that we should distinguish two types of confirmation measures for *x=>y* or *e*→*h*. One is to stand for the necessity of *x* compared with *x*_0_; the other is for the inevitability of *y*. *P*(*y*|*x*) may be good for the latter but not for the former. The former should be independent of *P*(*x*) and *P*(*y*). *P_d_* is such a one.

However, there is a problem with *P_d_*. If *P_d_* is 0 when *y* is uncorrelated to *x*, then *P_d_* should be negative instead of 0 when *x* inversely affects *y* (e.g., vaccine affects infection). Therefore, we need a confirmation measure between −1 and 1 instead of a probability measure between 0 and 1.

## 3. Methods

### 3.1. Defining Causal Posterior Probability

To avoid treating association as causality, we first explain what kind of posterior probabilities indicate causality. Posterior probability and conditional probability are often regarded as the same. However, Rubin emphasizes that probability *P*(*y^x^*) is not conditional; it is still marginal. To distinguish *P*(*y^x^*) and marginal probability *P*(*y*), we call *P*(*y^x^*), i.e., *P*(*y*|do(*x*)), the Causal Posterior Probability (CPP). What posterior probability is the CPP? We use the following example to explain.

About the population age distribution, let *z* be age and the population age distribution be *p*(*z*). We may define that a person with *z ≥* 60 is called an elderly; that is, *P*(*y*_1_|*z*) = 1 for *z* ≥ *z*_0_. The label of an elderly is *y*_1_, and the label of a non-elderly is *y*_0_. The probability of the elderly is:(18)$$P(y_{1})=\sum_{\mathrm{all}\;z}p(z)P(y_{1}|z)=\sum_{z\geq 60}p(z)$$

Let *x*_1_ denote the improved medical condition. After a period, *p*(*z*) becomes *p*(*z^x^*^1^) = *p*(*z*|do(*x*_1_)) and *P*(*y*_1_) becomes:(19)$$P(y_{1}{}^{x_{1}})=P(y_{1}|\mathrm{do}(x_{1}))=\sum_{z\geq 60}p(z|\mathrm{do}(x_{1})),$$

Let *x*_0_ be the medical condition existing already. We have:(20)$$P(y_{1}|\mathrm{do}(x_{0}))=\sum_{z\geq 60}p(z|\mathrm{do}(x_{0})).$$

There are similar examples:About whether a drug (*x*_1_) can lower blood pressure, blood sugar, blood lipid, or uric acid (*z*) or not, if *z* drops to a certain level *z*_0_, we say that the drug is effective (*y*_1_).About whether a fertilizer (*x*_1_) can increase grain yield (*z*), if *z* increases to a certain extent *z*_0_, the grain yield is regarded as a bumper harvest (*y*_1_).Can a process *x*_1_ reduce the deviation *z* of a product’s size? If the deviation is smaller than the tolerance (*z*_0_), we consider the product qualified (*y*_1_).

From the above examples, we can find that the action *x* can be the cause of a causal relationship because it can cause the change of probability distribution *p*(*z*) of objective result *z*, rather than the change of probability distribution *P*(*y*|∙) of outcome *y*. The reason is that *P*(*y*|∙) also changes with the dividing boundary *z*_0_. For example, if the dividing boundary of the elderly changes from *z*_0_ = 60 to *z*_0_′ = 65, the posterior probability *P*(*y*_1_|*z*_0_′) of *y*_1_ will become smaller. This change seemingly also reflects causality. However, the author thinks this change is due to a mathematical cause, which does not reflect the causal relationship we want to study. Therefore, we need to define the CPP more specifically.

**Definition** **1.** *Random variable Z takes a value z∈ {z_1_, z_2_, …} and p(z) is the probability distribution of the objective result. Random variable Y takes a value y∈{y_0_, y_1_} and represents the outcome, i.e., the classification label of z. The cause or treatment is x ∈ {x_0_, x_1_} or {x_1_, x_2_}. If replacing x_0_ with x_1_ (or x_1_ with x_2_) can cause the change of probability distribution p(z), we call x the cause, p(z|x) or p(z^x^) the CPP distribution, and P(y^x^) = P(y|do(x)) the CPP*.

According to the above definition, given *y*_1_, the conditional probability distribution *p*(*z*|*y*_1_) is not the CPP distribution because the probability distribution of *z* does not change with *y*.

Suppose that *x*_1_ is the vaccine for COVID-19, *y*_1_ is the infection, and *e*_1_ is the test-positive. Then *P*(*y*_1_|*x*_1_) or *P*(*y*_1_|do(*x*_1_)) is the CPP, whereas *P*(*y*_1_|*e*_1_) is not. We may regard *y*_1_ as the conclusion obtained by the best test, *e*_1_ is from a common test, and *P*(*y*_1_|*e*_1_) is the probability prediction of *y*_1_. *P*(*y*_1_|*e*_1_) is not a CPP because *e*_1_ does not change *p*(*z*) and the conclusion from the best test.

### 3.2. Using x_2_/x_1_ => y_1_ to Compare the Influences of Two Causes on an Outcome

In associated relationships, *x*_0_ is the negation of *x*_1_; they are complementary. However, in causal relationships, *x*_1_ is the substitute for *x*_0_. For example, consider taking medicines to cure the disease. Let *x*_0_ denote taking nothing, and *x*_1_ and *x*_2_ represent taking two different medicines. Each of *x*_1_ and *x*_2_ is a possible alternative to *x*_0_ instead of the negation of *x*_0_. Furthermore, in some cases, *x*_1_ may include *x*_0_ (see Section 4.3).

When we compare the effects of *x*_2_ and *x*_1_, it is unclear to use “*x*_2_ => *y*_1_” to indicate the causal relationship. Therefore, the author suggests that we had better replace “*x*_2_ => *y*_1_” with “*x*_2_/*x*_1_ => *y*_1_”, which means “replacing *x*_1_ with *x*_2_ will arise or increase *y*_1_”.

There are two reasons for using “*x*_2_/*x*_1_”:One is to express symmetry (*Cc*(*x*_2_/*x*_1_ => *y*_1_) = − *Cc*(*x*_1_/*x*_2_ => *y*_1_)) conveniently.Another is to emphasize that *x*_1_ and *x*_2_ are not complementary but alternatives for eliminating Simpson’s Paradox easily.

To compare *x*_1_ with *x*_0_, we may selectively use “*x*_1_/*x*_0_ => *y*_1_” or “*x*_1_ => *y*_1_”.

For Example 2 with a confounder, if we consider the treatment as replacing *x*_2_ with *x*_1_ in our imagination, we can easily understand why the number of patients in each group should be unchanged, that is, *P*(*g*|*x*_1_) = *P*(*g|x*_2_) = *P*(*g*). The reason is that the replacement will not change everyone’s kidney stone size.

In Example 3, *u* is a mediator, and the number of people in each group (with high or low blood pressure) is also affected by taking an antihypertensive drug *x*_1_. When we replace *x*_0_ with *x*_1_, *P*(*g*|*x*_1_) ≠ *P*(*g*|*x*_0_) ≠ *P*(*g*) is reasonable, and hence the weighting coefficients need not be adjusted. In this case, we can directly let *P*(*y*_1_|do(*x*)) = *P*(*y*_1_|*x*).

### 3.3. Deducing Causal Confirmation Measure Cc by the Methods of Semantic Information and Cross-Entropy

We use *x*_1_ => *y*_1_ as an example to deduce the causal confirmation measure *Cc*. If we need to compare any two causes, *x_i_* and *x_k_*, we may assume that one is default as *x*_0_.

Let *s*_1_ = “*x*_1_ => *y*_1_” and *s*_0_ = “*x*_0_ => *y*_0_”. We suppose that *s*_1_ includes a believable part with proportion *b*_1_ and a disbelievable part with proportion *b*_1_′. Their relation is *b*_1_′ + |*b*_1_| = 1. First, we assume *b*_1_ > 0; hence *b*_1_ = 1 − *b*_1_′. The two truth values of *s*_1_ are *T*(*s*_1_|*x*_1_) and *T*(*s*_1_|*x*_0_), as shown in the last row of Table 2.

Figure 4 shows how truth function *T(s*_1_|*x*) is related to *b*_1_ and *b*_1_′ for *b*_1_ > 0. *T*(*s*_1_|*x*_1_) = 1 means that example (*x*_1_, *y*_1_) makes *s*_1_ fully true; *T*(*s*_1_|*x*_0_) = *b*_1_′ is the truth value and the degree of disbelief of *s*_1_ for given counterexample (*x*_0_, *y*_1_).

The degree of belief optimized by a sampling distribution with the maximum semantic information or minimum cross-entropy criterion is the degree of causal confirmation, denoted by *Cc*_1_ = *Cc*(*x*_1_/*x*_0_ = >*y*_1_) = *b*_1_*.

The logical probability of *s*_1_ is (see Equation (7)):*T*(*s*_1_) = *P*(*x*_1_) + *P*(*x*_0_) *b*_1_′,(21) The predicted probability of *x*_1_ by *y*_1_ and *s*_1_ is:(22)$$P(x_{1}|\theta_{1})=\frac{T(s_{1}|x_{1})P(x_{1})}{T(s_{1})}=\frac{P(x_{1})}{P(x_{1})+P(x_{0}){b_{1}}^{\prime }},$$
where *θ_j_* can be regarded as the parameter of truth function *T*(*s_j_*|*x*).

The average semantic information conveyed by *y*_1_ and *s*_1_ about *x* is:(23)$$I(X;\theta_{1})=\sum_{i}P(x_{i}|y_{1})\log\frac{P(x_{i}|\theta_{1})}{P(x_{i})}=-\sum_{i}P(x_{i}|y_{1})\log P(x_{i})-H(X|\theta_{1}),$$
where *H(X|θ*_1_) is a cross-entropy. We suppose that sampling distribution *P*(*x, y*) has be modified so that *P*(*y*|*x*) = *P*(*y*|do(*x*)). According to the property of cross-entropy, *H(X|θ*_1_) reaches its minimum so that *I(X*; *θ_j_*) reaches its maximum as *P*(*x*|*θ*_1_) = *P*(*x|y*_1_), i.e.,
(24)$$\begin{array}{l} P(x_{0}|\theta_{1})=\frac{P(x_{0}){b_{1}}^{\prime }}{P(x_{1})+P(x_{0}){b_{1}}^{\prime }}=P(x_{0}|y_{1}),\;\\ P(x_{1}|\theta_{1})=\frac{P(x_{1})}{P(x_{1})+P(x_{0}){b_{1}}^{\prime }}=P(x_{1}|y_{1}). \end{array}$$ From the above two equations, we obtain:(25)$$\frac{P(x_{0})}{P(x_{1})}b_{1}{}^{\prime }=\frac{P(x_{0}|y_{1})}{P(x_{1}|y_{1})}.$$ Order:(26)$$m(x_{i},y_{j})=\frac{P(y_{j}|x_{i})}{P(y_{j})}=\frac{P(x_{i},y_{j})}{P(x_{i})P(y_{j})},i=0,1;\;j=0,1,$$
which represents the degree of correlation between *x_i_* and *y_j_* and may be independent of *P*(*x*) and *P*(*y*), unlike *P*(*x_i_, y_j_*). From Equations (25) and (26), we obtain the optimized degree of disbelief, i.e., the degree of disconfirmation:
*b_1_′** = *m*(*x*_0_,*y*_1_)/*m*(*x*_1_,*y*_1_). (27)
 Then we have the degree of confirmation of *s*_1_:(28)$$b_{1}*=1-b_{1}{}^{\prime }*=1-\frac{m(x_{0},y_{1})}{m(x_{1},y_{1})}=\frac{m(x_{1},y_{1})-m(x_{0},y_{1})}{m(x_{1},y_{1})}.$$ In the above formulas, we assume *b*_1_*> 0 and hence *m*(*x*_1_, *y*_1_) ≥ *m*(*x*_0_,*y*_1_). If *m*(*x*_1_, *y*_1_) < *m*(*x*_0_, *y*_1_), *b*_1_* should be negative, and *b*_1_′* should be *m(x*_1_, *y*_1_) / *m(x*_0_, *y_0_*). Then we have:(29)$$b_{1}*=-(1-b_{1}{}^{\prime }*)=-(1-\frac{m(x_{1},y_{1})}{m(x_{0},y_{1})})=\frac{m(x_{1},y_{1})-m(x_{0},y_{1})}{m(x_{0},y_{1})}.$$ Combining the above two equations, we derive the confirmation measure:(30)$$Cc(x_{1}=>y_{1})=b_{1}*=\frac{m(x_{1},y_{1})-m(x_{0},y_{1})}{\max(m(x_{1},y_{1}),m(x_{0},y_{1}))}.$$ Since $P(y_{j}|x_{i})=m(x_{i},y_{j})P(y_{j}),$ we also have:*b*_1_′* = *P*(*x*_0_|*y*_1_)/*P*(*x*_1_|*y*_1_), (31)
(32)$$Cc(x_{1}=>y_{1})=b_{1}*=\frac{P(y_{1}|x_{1})-P(y_{1}|x_{0})}{\max(P(y_{1}|x_{1}),P(y_{1}|x_{0}))}=\frac{R-1}{\max(R,1)},$$
where *R* = *P*(*y*_1_|*x*_1_) / *P*(*y*_1_|*x*_0_) is the relative risk or the likelihood ratio used for *P_d_*.

Measure *Cc* has the normalizing property since its maximum is 1 as *m*(*x*_0_, *y*_1_) = 0 and the minimum is −1 as *m*(*x*_1_, *y*_1_) = 0. It has cause symmetry since:(33)$$\begin{array}{ll} Cc(x_{0}/x_{1}=>y_{1}) & =\frac{m(x_{0},y_{1})-m(x_{1},y_{1})}{\max(m(x_{0},y_{1}),m(x_{1},y_{1}))}\\ & =-\frac{m(x_{1},y_{1})-m(x_{0},y_{1})}{\max(m(x_{1},y_{1}),m(x_{0},|y_{1}))}=-Cc(x_{1}/x_{0}=>y_{1}). \end{array}$$

Similarly, letting probability distribution *P*(*y*|*x*_1_) be the linear combination of a uniform probability distribution and a 0–1 distribution, we can obtain another causal confirmation measure:(34)$$Ce(x_{1}=>y_{1})=\frac{P(y_{1}|x_{1})-P(y_{0}|x_{1})}{\max(P(y_{1}|x_{1}),P(y_{0}|x_{1}))}=\frac{2P(y_{1}|x_{1})-1}{\max(P(y_{1}|x_{1}),1-P(y_{1}|x_{1}))}.$$ This measure can be regarded as the direct extension of Bayesian confirmation measure *c**(*e*_1_→*h*_1_) [8]. It increases monotonically with the Bayesian confirmation measure *f*(*h*_1_, *e*_1_) = *P*(*h*_1_|*e*_1_), which is used by Fitelson et al. [5,32]. However, *Ce* has the normalizing property and the outcome symmetry:*Ce*(*x*_1_ => *y*_1_) = − *Ce*(*x*_1_ => *y*_0_).(35)

### 3.4. Causal Confirmation Measures Cc and Ce for Probability Predictions

From *y*_1_, *b*_1_*, and *P*(*x*), we can make the probability prediction about *x*:(36)$$P(x_{1}|\theta_{1})=\frac{P(x_{i})}{P(x_{1})+b_{1}{}^{\prime }*P(x_{0})},\;P(x_{0}|\theta_{1})=\frac{P(x_{0})b_{1}{}^{\prime }*}{P(x_{1})+b_{1}{}^{\prime }*P(x_{0})},$$
where *b*_1_* > 0, *θ*_1_ represents *y*_1_ with *b*_1_′*, and *θ*_0_ means *y*_0_ with *b*_0_′*. If *b_1_**< 0, we let *T(s*_1_|*x*_1_) = *b*_1_′ and *T(s*_1_|*x*_0_) = 1, and then use the above formula.

Following the probability prediction with Bayesian confirmation measure *c** [8], we can also make probability prediction for given *x*_1_ and ***C****e*_1_. For example, when *Ce*_1_ is greater than 0, there is:(37)$$P(y_{1}|\theta_{x1})=1/(2-Ce_{1}),$$
where *θ_x_*_1_ denotes *x*_1_ and *Ce*_1_.

Given the semantic channel ascertained by *b*_1_ > 0 and *b*_0_ > 0, as shown in Table 2, we can obtain the corresponding Shannon channel *P*(*y*|*x*). According to Equation (32), we can deduce:(38)$$\begin{array}{l} P(y_{1}|x_{1})=\frac{1-b_{0}{}^{\prime }}{1-b_{1}{}^{\prime }b_{0}{}^{\prime }},\;P(y_{0}|x_{0})=\frac{1-b_{1}{}^{\prime }}{1-b_{1}{}^{\prime }b_{0}{}^{\prime }},\\ P(y_{0}|x_{1})=1-P(y_{1}|x_{1}),\;P(y_{1}|x_{0})=1-\;P(y_{0}|x_{0}). \end{array}$$

## 4. Results

### 4.1. A Real Example of Kidney Stone Treatments

Table 3 shows Example 2 with detailed data about kidney stone treatments [15]. The data were initially provided in [16]. In Table 3, *% means a success rate, and the number behind it is the number of patients. The stone size is a confounder. The conclusion from every group (with small or large stones) is that treatment *x*_2_ (i.e., treatment A in [15]) is better than treatment *x*_1_ (i.e., treatment B in [15]); whereas the conclusion according to average success rates, *P*(*y*_1_*|x*_2_) = 0.78 and *P*(*y*_1_|*x*_1_) = 0.83, treatment *x*_1_ is better than treatment *x*_2_. There seems to be a paradox.

We used *P*(*g*) instead of *P*(*g*|*x*_1_) or *P*(*g|x*_2_) as the weighting coefficient for *P*(*y*_1_|do(*x*_1_)) and *P*(*y*_1_|do(x_2_)). After replacing *P*(*y*_1_|*x*_1_) with *P*(*y*_1_|do(*x*_1_)) and *P*(*y*_1_|*x*_2_) with *P*(*y*_1_|do(*x*_2_)), we derived *Cc*_1_ = *Cc*(*x*_2_/*x*_1_ => *y*_1_) = 0.06 (see Table 3), which means that the overall conclusion is that treatment *x*_2_ is better than treatment *x*_1_.

For *Cc*_1_ in Table 3, we used treatment *x*_1_ as the default; the degree of causal confirmation *Cc*_1_
*= Cc*(*x*_2_/*x*_1_ => *y*_1_) is 0.06. If we used *x*_2_ as the default, *Cc*_1_ = *Cc*(*x*_1_/x_2_ => *y*_1_) = −0.06. Using measure *Cc*, we need not worry about which of *P*(*y*_1_|do(*x*_1_)) and *P*(*y*_1_|do(*x*_2_)) is larger, whereas, using *P_d_*, we have to consider that before calculating *P_d_*.

We used the incremental school’s confirmation measure *D*(*x*_1_, *y*_1_) to compare *x*_1_ and *x*_2_. We obtained:●*P*(*y*_1_) = *P*(*x*_1_)*P*(*y*_1_|*x*_1_) + *P*(*x*_2_)*P*(*y*_1_*|x*_2_) = 0.805,●*P*(*y*_1_*|x*_2_, *g*_1_) − *P*(*y*_1_) = 0.93 − 0.805 > 0,●*P*(*y*_1_*|x*_2_, g_2_) − *P*(*y*_1_) = 0.73 − 0.805 < 0, and●*D*(*x*_1_, *y*_1_) = *P*(*y*_1_*|x*_1_) − *P*(*y*_1_) = 0.83 − 0.805 > 0●*D(x_2_, y_2_) = P*(*y*_1_*|x*_2_) − *P*(*y*_1_) = 0.78 − 0.805 < 0.

The results mean that *x*_1_ is better than *x*_2_. There seems to be no paradox only because the paradox is avoided rather than eliminated when we use *D*(*x*_1_, *y*_1_).

We tested Equation (38) by the aforementioned example. The Shannon channel *P*(*y*|*x*) derived from the two degrees of disconfirmation *b*_1_′* and *b*_0_′* is the same as *P*(*y*|do(*x*)) shown in the last two rows of Table 3.

### 4.2. An Example of Eliminating Simpson’s Paradox with COVID-19

Table 4 shows Example 2 with detailed data about the CFRs of COVID-19. The original data were obtained from the website of the Centers for Disease Control and Prevention (CDC) in the United States up until 2 July 2022 [33]. The data only include reported cases; otherwise, the CFRs should be lower. The *x*_1_ represents the non-Hispanic white and *x*_2_ means the other races. *P(y*_1_*|x*_1_*, g*) and *P(y*_1_*|x*_2_*, g*) are the CFRs of *x*_1_ and *x*_2_ in an age group *g*. See Appendix A for the original data and median results.

Table 5 shows that the overall (average) CFRs vary before and after we change the weighting coefficient from *P(g|x*) to *P(g*).

From Table 4, we can find that for different age groups, the CFR of the non-Hispanic whites is lower than or close to that of the other races. However, for all age groups (see Table 5), the overall (average) CFR (1.04) of the non-Hispanic whites is higher than the CFR (0.73) of the other races. After replacing *P*(*g|x*) with *P*(*g*), the overall CFR (0.80) of the non-Hispanic whites is also lower than that (1.05) of the other races.

We followed Fitelson to use *D*(*x*_1_, *y*_1_) to assess the risk. The average CFR is 0.97 (found on the same website [33]). We obtained:*D*(non-Hispanic whites, death) = *P*(*y*_1_|*x*_1_) − *P*(*y*_1_) = 1.04 − 0.97 = 0.07,
*D*(other people, death) = *P*(*y*_1_*|x*_2_) − *P*(*y*_1_) = 0.73 − 0.97 = −0.14,
which means that non-Hispanic whites are at higher risk.

### 4.3. COVID-19: Vaccine’s Negative Influences on the CFR and Mortality

Using causal probability measure *P_d_* is not convenient to measure the “probability” of “vaccine => infection” or “vaccine => death”, since *P_d_* is regarded as the probability, whose minimum value is 0, while the vaccine’s influence is negative. However, there is no problem using *Cc* because *Cc* can be negative.

Table 6 shows data obtained from the website of the US CDC [34] and the two degrees of causal confirmation. The numbers of cases and deaths are among 100,000 people (ages over 5) in a week (from June 20 to 26, 2022).

The negative degree of causal confirmation −0.63 means that the vaccine reduced the infection rate by 63%. The −0.79 means that the vaccine reduced the CFR by 79%.

To know the impact of COVID-19 on population mortality, we need to compare the regular mortality rate due to common reasons (*x*_0_) with the new mortality rate due to common reasons plus COVID-19 (*x*_1_) during the same period (such as one year). Since the average lifespan of people in the United States is 79 years old, the annual mortality rate is about 1/79 = 0.013. From Table 6, we can derive that the yearly mortality rate caused by COVID-19 is 0.001 (for unvaccinated people) or 0.00018 (for vaccinated people).

People who died due to COVID-19 may also die in the same year for common reasons. Therefore, the new mortality rate should be less than the sum of the two mortality rates. Assume that *x*_0_ and *x*_1_ are independent of each other. Then new mortality rate *P*(*y*_1_|*x*_1_) should be 0.013 + 0.001 − 0.013 × 0.001 ≈ 0.014 (for unvaccinated people) or 0.013 + 0.00018 − 0.013 × 0.00018 ≈ 0.01318 (for vaccinated people). Table 7 shows the degree of causal confirmation of COVID-19 leading to mortality, for which we assume *P*(*y*_1_|*x*) = *P*(*y*_1_|do(*x*)).

In the last line, *Cc*_1_ = 0.07 means that among unvaccinated people who die, 7% are due to COVID-19. Moreover, *Cc* = 0.014 means that among the vaccinated people who die, 1.4% are due to COVID-19.

If we used *x*_1_ = COVID-19 instead of *x*_1_ = *x*_0_ + COVID-19, we would get a strange conclusion that COVID-19 could reduce deaths.

We obtained the above results without considering the vaccine’s side effects, possibly resulting in chronic deaths.

## 5. Discussion

### 5.1. Why Can P_d_ and Cc Better Indicate the Strength of Causation Than D in Theory?

We call *m*(*x_i_, y_j_*) (*i* = 0,1; *j* = 0,1) the probability correlation matrix, which is not symmetrical. Although there exists *P*(*x, y*) first and then *m*(*x, y*) from the perspective of calculation, there exists *m*(*x, y*) first and then *P*(*x, y*) from the perspective of existence. That is, given *P*(*x*), *m*(*x, y*) only allows specific *P*(*y*) to happen.

We can also make probability predictions with *m*(*x, y*) (like using Bayes’ formula):(39)$$\begin{array}{l} P(y|x_{1})=P(y)m(x_{1},y)/m(x_{1}),m(x_{1})=\sum_{y}P(y)m(x_{1},y),\\ P(x|y_{1})=P(x)m(x,y_{1})/m(y_{1}),m(y_{1})=\sum_{x}P(x)m(x,y_{1}). \end{array}$$

From Equations (27)–(30), we can find that *P_d_* and *Cc* only depend on *m*(*x, y*) and are independent of *P*(*x*) and *P*(*y*). The two degrees of disconfirmation, *b*_1_′* and *b*_0_′*, ascertain a semantic channel and a Shannon channel. Therefore, the two degrees of causal confirmation, *Cc*_1_ = *b*_1_* and *Cc*_0_ = *b*_0_*, indicate the strength of the constraint relationship (causality) from *x* to *y*. Like *Cc*, measure *P_d_* is also only related to *m*(*x, y*). *D* and Δ**P_x_^y^* are different; they are related to *P*(*x*), so they do not indicate the strength of causation well.

For example, considering the vaccine’s effect on the CFR of COVID-19 (see Table 7), *P_d_* or *Cc* are irrelated to vaccination coverage rate *P*(*x*_1_), whereas measure Δ**P_x_^y^* is related to *P*(*x*_1_). Measure *D* is associated with *P*(*y*) and is also related to *P*(*x*_1_). *P_d_* and *Cc*_1_ obtained from one region also fits other areas for the same variant of COVID-19. In contrast, Δ**P_x_^y^* and *D* are not universal because the vaccination coverage rate *P*(*x*_1_) differs in different areas.

According to the incremental school’s view of Bayesian confirmation, *P*(*y*_1_) is a prior probability, and *P*(*y*_1_|*x*) − *P*(*y*_1_) is its increment. However, when measure *D* is used for causal confirmation, *P*(*y*_1_) is obtained from *P*(*x*) and *P*(*y*_1_|*x*) after the treatment, so *P*(*y*) is no longer a priori probability, which is also a fatal problem with the incremental school.

In addition, as the result of induction, *Cc* and *P_d_* can indicate the degree of belief of a fuzzy major premise and can be used for probability predictions, whereas *D* and Δ**P_x_^y^* cannot.

### 5.2. Why Are P_d_ and Cc Better than D in Practice?

Two calculation examples in Section 4.1 and Section 4.2 support the conclusion that measures *P_d_* and *Cc* are better than *D* in practice. The reasons are as follows.

#### 5.2.1. *P_d_* and *Cc* Have Precise Meanings in Comparison with *D*

*Cc*_1_ = *Cc* (*x*_1_/*x*_0_ => *y*_1_) indicates what percentage of the result *y* is due to *x*_1_ instead of *x*_0_. For example, Table 6 shows that according to the virulence of the virus, COVID-19 will increase the mortality rate of vaccinated people from 1.3% to 1.318%. Therefore, the degree of causal confirmation is *Cc*_1_ = *P_d_* = 0.014, which means that 1.4% of the deaths will be due to COVID-19. However, the meanings of *D* and Δ**P_x_^y^* are not precise.

Different from measure *RD* (see Equation (1)), *P_d_* and *Cc* indicate relative risk or the relative change of the outcome. Many people think COVID-19 is very dangerous because it can kill millions in a country. However, the mortality rate it brings is much lower than that caused by common reasons. *P_d_* and *Cc* can reveal the relative change in the mortality rate (see Table 7). Although it is essential to reduce or delay deaths, it is also vital to decrease the economic loss due to the fierce fight against the pandemic. Therefore, *P_d_* and *Cc* can help decision-makers balance between reducing or delaying deaths and reducing financial losses.

#### 5.2.2. The Confounder’s Effect Is Removed from *P_d_* and *Cc*

When there is a confounder, as shown in Section 4.1, using *P_d_* or *Cc*, we can eliminate Simpson’s Paradox and make the overall conclusion consistent with the grouping conclusion: treatment *x*_2_ is better than treatment *x*_1_. For example, if we use *D* to compare the success rates of two treatments, although we can avoid Simpson’s Paradox, the conclusion is unreasonable. The reason is that we neglect the difficulties of treatments for different sizes of kidney stones. If a hospital only accepts patients who are easy to treat, its overall success rate must be high; however, such a hospital may not be a good one.

#### 5.2.3. *P_d_* and *Cc* Allow Us to View the Third Factor, *u*, from Different Perspectives

For the example in Section 4.2, if we think that one’s longevity is related to one’s race, we can take the lifespan as a mediator and then directly accept the overall conclusion (non-Hispanic whites have a higher CFR than other people). On the other hand, if we believe that one’s longevity is not due to one’s race, then the lifespan is a confounder. Therefore, we can make the overall conclusion consistent with the grouping conclusion, and then use *P_d_* and *Cc*.

It is worth noting that it is concluded that the CFR of non-Hispanic whites is lower than that of other people, probably because medical conditions affect the CFRs. However, existing data do not contain information about the medical conditions of different races. Otherwise, the CFRs of different races might be similar if we use the medical condition as a confounder. This issue is worth researching further.

### 5.3. Why Is It Better to Replace P_d_ with Cc?

Section 4.3 provides the calculation of the two negative degrees of causal confirmation that reflect the impacts of the vaccine on infection and mortality. The negative degrees of confirmation mean that the vaccine can reduce illnesses and deaths. However, if we use *P_d_* as the probability of causation, *P_d_* can only take its lower limit 0. Although we can replace *P_d_*(vaccinated => death) with *P_d_*(unvaccinated => death) to ensure *P_d_* > 0, it does not conform to our thinking habits to take being vaccinated as the default cause. In addition, *Cc* has cause symmetry, whereas *P_d_* does not.

When we used *P_d_* to compare two causes *x*_1_ and *x*_2_, such as two treatments for kidney stones (see Section 4.1), we had to consider which of *P*(*y*_1_*|x*_2_) and *P*(*y*_1_|*x*_1_) was larger. However, using *Cc*, we needed to not consider that because it is unnecessary to worry about if (*R* − 1)/*R* < 0.

The correlation coefficient in mathematics is between 1 and −1. *Cc* can be understood as the probability correlation coefficient. The difference is that the former has only one coefficient between *x* and *y*, whereas the latter has two coefficients: *Cc*_1_ = *Cc*(*x*_1_ => *y*_1_) and *Cc*_0_ = *Cc*(*x*_0_ => *y*_0_).

### 5.4. Necessity and Sufficiency in Causality

Measures *P_d_* and *Cc* only indicate the necessity of cause *x* to outcome *y*; they do not reflect the sufficiency of *x* or the inevitability of *y*. On the other hand, measure *f* = *P*(*y*|*x*) and *Ce* can indicate the outcome’s inevitability.

The medical industry uses the odds ratio to indicate both the necessity and sufficiency of the cause to the outcome. The odds rate [2] is:(40)$$OR=\frac{P(y_{1}|x_{1})}{P(y_{1}|x_{0})}\times \frac{P(y_{0}|x_{0})}{P(y_{0}|x_{1})}.$$

It is the product of two likelihood ratios. We can use:(41)$$OR_{N}=\frac{OR-1}{\max(OR,1)}$$
as the confirmation measure of both *x*_0_ => *y*_0_ and *x*_1_ => *y*_1_ for the same purpose. However, *OR_N_* has the normalizing property and symmetry.

### 5.5. The Relationship between Bayesian Confirmation Measures b* and c*, and Causal Confirmation Measures Cc and Ce

Suppose that *P*(*y*_1_|*x*) has been modified for *P*(*y*_1_|*x*) = *P*(*y*_1_|do(*x*)). Causal confirmation measure *Cc* is equal to channels’ confirmation measure *b** [8] in value, i.e.,
*Cc*(*x*_1_ => *y*_1_) = [*P*(*y*_1_|*x*_1_) − *P*(*y*_1_|*x*_0_)]/max(*P*(*y*_1_|*x*_1_), *P*(*y*_1_|*x*_0_)) = *b**(*y*_1_→*x*_1_). (42) However, their antecedents and consequents are inverted, which means that if *x*_1_ is the cause of *y*_1_, then *y*_1_ is the evidence of *x*_1_. For example, if COVID-19 infection is the cause of the test-positive, then the test-positive is the evidence of the infection.

Causal confirmation measure *Ce* indicating the inevitability of the outcome is equal to prediction confirmation measure *c**(*x*_1_→*y*_1_) in value, i.e.,
*Ce*(*x*_1_ => *y*_1_) = [*P*(*y*_1_|*x*_1_) − *P*(*y*_0_|*x*_1_)]/max(*P*(*y*_1_|*x*_1_), *P*(*y*_0_|*x*_1_)) = *c**(*x*_1_→*y*_1_). (43)

Their antecedents and consequents are the same.

However, from the right sides’ values of the above two equations, we may not be able to obtain the left sides’ values because an associated relationship may not be a causal relationship.

## 6. Conclusions

Fitelson, a representative of the incremental school of Bayesian confirmation, used *D*(*x*_1_, *y*_1_) = *P*(*y*_1_|*x*_1_) − *P*(*y*_1_) to denote the supporting strength of the evidence to the consequence and extended this measure for causal confirmation without considering the confounder. This paper has shown that measure *D* is incompatible with the ECIT and popular risk measures, such as *P_d_* = max(0, (*R* − 1)/*R*). Using *D*, one can only avoid Simpson’s Paradox but he cannot eliminate it or provide a reasonable explanation as the ECIT does.

On the other hand, Rubin et al. used *P_d_* as the probability of causation. *P_d_* is better than *D*, but it is improper to call *P_d_* a probability measure and use the probability measure to measure causation. If we use *P_d_* as a causal confirmation measure, it lacks the normalizing property and symmetry that an ideal confirmation measure should have.

This paper has deduced causal confirmation measure *Cc*(*x*_1_ => *y*_1_) = (*R* – 1) / max(*R*, 1) by the semantic information method with the minimum cross-entropy criterion. *Cc* is similar to the inductive school’s confirmation measure *b** proposed by the author earlier. However, the positive examples’ proportion *P*(*y*_1_|*x*_1_) and the counterexamples’ proportion *P*(*y*_1_|*x*_0_) are replaced with *P*(*y*_1_|do(*x*_1_)) and *P*(*y*_1_|do(*x*_0_)) so that *Cc* is an improved *P_d_*. Compared with *P_d_*, *Cc* has the normalizing property (it changes between –1 and 1) and the cause symmetry (*Cc*(*x*_0_/*x*_1_ => *y*_1_) = −*Cc* (*x*_1_/*x*_0_ => *y*_1_)). Since *Cc* may be negative, it is also suitable for evaluating the inhibition relationship between cause and outcome, such as between vaccine and infection.

This paper has provided some examples with Simpson’s Paradox for calculating the degrees of causal confirmation. The calculation results show that *P_d_* and *Cc* are more reasonable and meaningful than *D*, and *Cc* is better than *P_d_* mainly because *Cc* may be less than zero. In addition, this paper has also provided a causal confirmation measure *Ce*(*x*_1_ => *y*_1_) that indicates the inevitability of the outcome *y*_1_.

Since measure *Cc* and the ECIT support each other, the inductive school of Bayesian confirmation are also supported by the ECIT and the epidemical risk theory.

However, like all Bayesian confirmation measures, causal confirmation measure *Cc* and *Ce* also use size-limited samples, hence, the degrees of causal confirmation are not strictly reliable. Therefore, replacing a degree of causal confirmation with a degree interval is necessary to retain the inevitable uncertainty. This work needs further studies by combining existing theories.

## Figures and Tables

**Figure 1 entropy-25-00143-f001:**
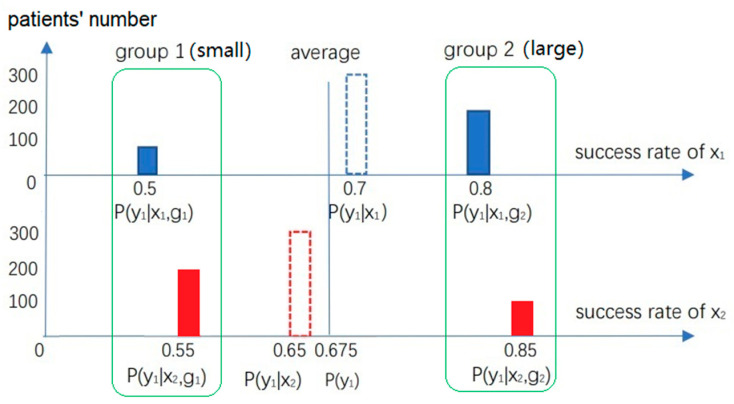
Illustrating Simpson’s Paradox. In each group, the success rate of *x*_2_, *P*(*y*_1_*|x*_2_, *g*), is higher than that of *x*_1_, *P*(*y*_1_|*x*_1_, *g*); however, using the method of finding the center of gravity, we can see that the overall success rate of *x*_2_, *P*(*y*_1_|*x*_2_) = 0.65, is lower than that of *x*_1_, *P*(*y*_1_|*x*_1_) = 0.7.

**Figure 2 entropy-25-00143-f002:**
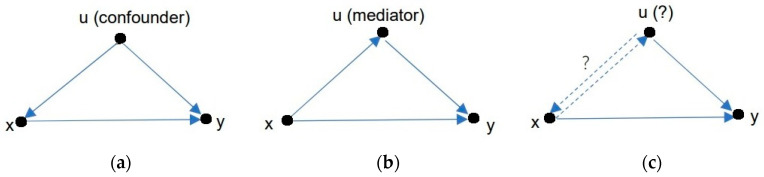
Three causal graphs: (**a**) for Example 2; (**b**) for Example 3; (**c**) for Examples 4 and 1.

**Figure 3 entropy-25-00143-f003:**
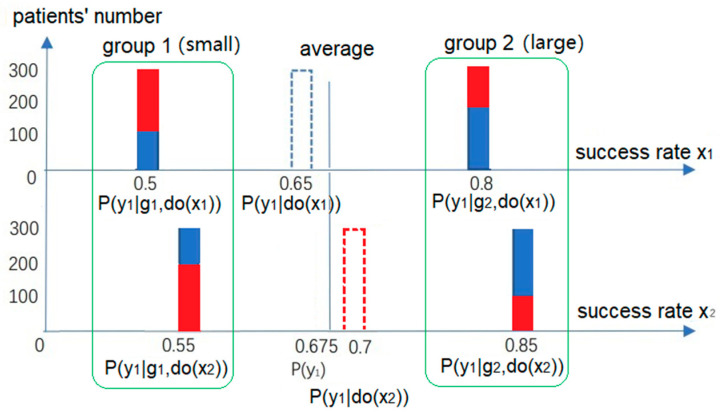
Eliminating Simpson’s Paradox as the confounder exists by modifying the weighting coefficients. After replacing *P*(*g*_k_*|x_i_*) with *P*(*g*_k_) (*k =* 1,2; *i* = 1,2), the overall conclusion is consistent with the grouping conclusion; the average success rate of *x*_2_, *P*(*y*_1_|do(*x*_2_)) = 0.7, is higher than that of *x*_1_, *P*(*y*_1_|do(*x*_1_)) = 0.65.

**Figure 4 entropy-25-00143-f004:**
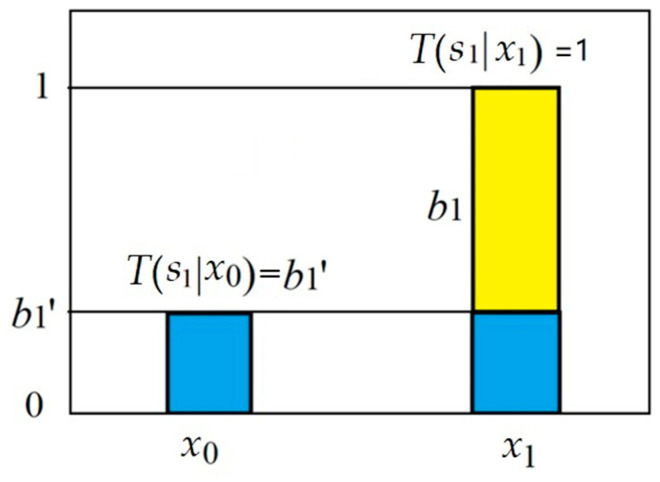
The truth function of *s*_1_ includes a believable part with proportion *b*_1_ and a disbelievable part with proportion *b*_1_′.

**Table 1 entropy-25-00143-t001:** Comparing *P_d_* and Δ**P_x_^y.^*

	*P*(*y*_1_|*x*_1_)	*P*(*y*_1_|*x*_0_)	*P_d_*	Δ**P_x_^y^*	Comparison
No big difference	0.9	0.8	0.11	0.5	*P_d_* << Δ**P_x_^y^*
No counterexample	0.2	0	1	0.2	*P_d_* >> Δ**P_x_^y^*

**Table 2 entropy-25-00143-t002:** The truth values of *s*_0_ = “*x*_0_ => *y*_0_” and *s*_1_ = “*x*_1_ => *y*_1_”.

	*T*(*s*|*x*_0_)	*T*(*s*|*x*_1_)
*s*_0_ = “*x*_0_ => *y*_0_”	1	*b*_0_′
*s*_1_ = “*x*_1_ => *y*_1_”	*b*_1_′	1

**Table 3 entropy-25-00143-t003:** Comparing two treatments’ success rates (*y*_1_ means the success).

	Treat. *x*_1_	Treat. *x*_2_	Number	*P*(*g*) or *Cc*
Small stones (*g*_1_)	87%/270	93%/87 *	357	0.51
Large stones (*g*_2_)	69%/80	73%/263	343	0.49
Overall	83%/350	78%/350	700	
*P*(*y*_1_|*x*)	0.83	0.78		[(*P*(*y*_1_|x_2_) − *P*(*y*_1_|*x*_1_)]/*P*(*y*_1_|x_2_) = −0.064
*P*(*y*_1_|do(*x*))	0.78	0.83		*Cc*_1_ = *Cc*(x_2_/*x*_1_ => *y*_1_) = 0.06
*P*(*y*_0_|do(*x*))	0.22	0.17		*Cc*_0_ = *Cc*(*x*_1_/*x*_2_ => *y*_0_) = 0.23

* “87%/270” means that the success rate is 87%, and the number in this subgroup is 270.

**Table 4 entropy-25-00143-t004:** The CFRs of COVID-19 of non-Hispanic white (*x*_1_) and other people (*x*_2_) from different age groups.

Age Group (*g*)	*P*(*x*_1_*|g*)	*P*(*g*)	*P*(*y*_1_|*x*_1_, *g*)	*P*(*g|x*_1_)	*P*(*y*_1_|*x*_2_, *g*)	*P*(*g|x*_2_)
0–4 Years	44.200	0.041	0.0002	0.0349	0.0002	0.0480
5–11 Years	44.200	0.078	0.0001	0.0659	0.0001	0.0907
12–15 Years	46.300	0.052	0.0001	0.0458	0.0001	0.0578
16–17 Years	48.700	0.029	0.0001	0.0268	0.0002	0.0307
18–29 Years	48.700	0.223	0.0004	0.2081	0.0006	0.2388
30–39 Years	49.300	0.178	0.0011	0.1681	0.0019	0.1883
40–49 Years	51.000	0.146	0.0030	0.1427	0.0048	0.1493
50–64 Years	59.100	0.163	0.0102	0.1843	0.0144	0.1389
65–74 Years	67.300	0.055	0.0333	0.0704	0.0457	0.0373
75–84 Years	72.900	0.025	0.0762	0.0356	0.0938	0.0144
85+ Years	76.300	0.012	0.1606	0.0173	0.1751	0.0059
sum		1		1		1

**Table 5 entropy-25-00143-t005:** Comparing the CFRs of non-Hispanic whites and the other people.

	The CFR of Non-Hispanic Whites (*x*_1_)	The CFR of of Other People (*x*_2_)	Risk Measure *
*P*(*y*_1_|*x*)	1.04	0.73	*P_d_* = (*R* − 1)/*R* = 0.30
*P*(*y*_1_|do(*x*))	0.80	1.05	*P_d_* = 0; *Cc*(*x*_1_/*x*_2_=>*y*_1_) = −0.28

* *R* = *P*(*y*_1_|*x*_1_)/*P*(*y*_1_*|x*_2_).

**Table 6 entropy-25-00143-t006:** The negative degrees of causal confirmation for accessing that the vaccine affects infections and deaths.

	Unvaccinated (*x*_0_)	Vaccinated (*x*_1_)	*Cc*
Cases	512.6	189.5	*Cc*(*x*_1_/*x*_0_ => *y*_1_) = −0.63
Deaths	1.89	0.34	*Cc*(*x*_1_/*x*_0_) => *y*_1_) = −0.79
Mortality rate	0.001	0.00018	

**Table 7 entropy-25-00143-t007:** Using *Cc* to measure the impact of COVID-19 on the mortality rates.

	Mortality Rate *P*(*y*_1_|*x*)	Unvaccinated	Vaccinated
*x*_0_: common reasons	*P*(*y*_1_|*x*_0_)	0.013	0.013
*x*_1_: *x*_0_ plus COVID-19	*P*(*y*_1_|*x*_1_)	0.014	0.01318
*Cc*_1_ = *Cc*(*x*_1_/*x*_0_ => *y*_1_)		0.07	0.014

## Data Availability

Not applicable.

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
