# Peer review of "Causal Confirmation Measures: From Simpson’s Paradox to COVID-19"

_entropy, 2023, doi:10.3390/e25010143_

Round 1

Reviewer 1 Report

1.       Line 132: Can you elaborate more about P(y1|g, x1) < P(y1|g, x0)? And P(y1|.) means a higher probability of recovering health? Am I correct?

2.       Can you elaborate that what if m(x1, y1) < m(x0, y1)? The Table 2 would not make sense if b1 > 1?

3.       I’m curious to replace the table whether the author can provide more numerical examples for Table 4 into different groups of people (e.g. by underlying health conditions) to provide more insights or superior of the method in Bayesian way

4.       Following #3, how would you consider the underlying health conditions as a confounder or mediator? Can you provide more numerical results of this part?

Reviewer 2 Report

The manuscript addresses a quite important topic: causal confirmation measures. The manuscript provides a quite large discussion of the topic mainly centered on the previous contributions of the author.

Unfortunately the style and contents do not seem very suitable for a journal like Entropy. It seems to me that journals with a more theoretical/logical audience would be more appropriate. Moreover, the work has a clear overview character but does not provide a sufficiently broad background (just as a couple of examples, the fuzzy logic measures and beliefs are abruptly introduced in section 2). Several of these topics are not the typical background of Entropy audience, rendering the work very difficult for the typical reader. If the manuscript is to be published in Entropy, a major rewriting, to render the work more digestible, is indispensable. I have also to say that the innovative content of the work seems very limited and not very clear. This is another aspect that would probably advise publication in a different journal. I’ll leave to the editor whether to opt for a major revision or for a withdrawal and resubmission.

More detailed comments:

11)      The English needs polishing.

22)      Figure 1 needs improving: it is not obvious which the number of cases in each group is.

33)      The format of the formulas is not uniform along the document.

Round 2
